# Influence of Renal Dysfunction on the Differential Behaviour of Procalcitonin for the Diagnosis of Postoperative Infection in Cardiac Surgery

**DOI:** 10.3390/jcm11247274

**Published:** 2022-12-07

**Authors:** Olga de la Varga-Martínez, Marta Martín-Fernández, María Heredia-Rodríguez, Francisco Ceballos, Hector Cubero-Gallego, Juan Manuel Priede-Vimbela, Miguel Bardají-Carrillo, Laura Sánchez-de Prada, Rocío López-Herrero, Pablo Jorge-Monjas, Eduardo Tamayo, Esther Gómez-Sánchez

**Affiliations:** 1Department of Anaesthesiology, Infanta Leonor University Hospital, Gran Via del Este 80, 28031 Madrid, Spain; 2BioCritic, Group for Biomedical Research in Critical Care Medicine, Ramon y Cajal Ave. 7, 47005 Valladolid, Spain; 3Center for Biomedical Research in Infectious Diseases Network (CIBERINFEC), Carlos III Health Institute, 28029 Madrid, Spain; 4Department of Medicine, Faculty of Medicine, Universidad de Valladolid, 47005 Valladolid, Spain; 5Department of Anaesthesiology, Clinical University Hospital of Salamanca, P.° de San Vicente, 58, 37007 Salamanca, Spain; 6Viral Infection and Immunity Unit, National Center for Microbiology, Carlos III Health Institute, 28029 Madrid, Spain; 7Interventional Cardiology Unit, Cardiology Department, Hospital del Mar, 08003 Barcelona, Spain; 8Department of Anaesthesiology, Clinic University Hospital of Valladolid, Ramon y Cajal Ave. 3, 47003 Valladolid, Spain; 9Microbiology and Immunology Department, Hospital Clínico Universitario de Valladolid, 47003 Valladolid, Spain; 10Department of Surgery, Faculty of Medicine, Universidad de Valladolid, 47005 Valladolid, Spain

**Keywords:** procalcitonin, marker, postoperative infection, renal dysfunction

## Abstract

Background: procalcitonin is a valuable marker in the diagnosis of bacterial infections; however, the impairment of renal function can influence its diagnostic precision. The objective of this study is to evaluate the differential behaviour of procalcitonin, as well as its usefulness in the diagnosis of postoperative pulmonary infection after cardiac surgery, depending on the presence or absence of impaired renal function. Materials and methods: A total of 805 adult patients undergoing cardiac surgery with extracorporeal circulation (CBP) were prospectively recruited, comparing the behaviour of biomarkers between the groups with and without postoperative pneumonia and according to the presence or absence of renal dysfunction. Results: Pulmonary infection was diagnosed in 42 patients (5.21%). In total, 228 patients (28.32%) presented postoperative renal dysfunction. Procalcitonin was significantly higher in infected patients, even in the presence of renal dysfunction. The optimal procalcitonin threshold differed markedly in patients with renal dysfunction compared to patients without renal dysfunction (1 vs. 0.78 ng/mL *p* < 0.05). The diagnostic accuracy of procalcitonin increased significantly when the procalcitonin threshold was adapted to renal function. Conclusions: Procalcitonin is an accurate marker of postoperative infection in cardiac surgery, even in the presence of renal dysfunction. Renal function is an important determinant of procalcitonin levels and, therefore, its diagnostic thresholds must be adapted in the presence of renal dysfunction.

## 1. Introduction

After a major surgery such as cardiac surgery with cardiopulmonary bypass (CPB), postoperative pulmonary infection has been reported to occur in 5.7–21.6% of patients [1,2], leading to death in up to 31.9% of cases [3,4,5]. Postoperative pulmonary infection is often diagnosed late. This is caused, on the one hand, by several confounding factors due to the activation of inflammatory cascades, which may lead to systemic inflammatory response syndrome (SIRS) [6,7], or, on the other hand, by the poor diagnostic performance of current infection biomarkers in the postoperative period [1]. This growing concern gives rise to an increase in research studies looking for new biomarkers, such as LIFTS or long non-coding RNAs (lncRNAs), with the aim of improving the diagnosis and prognosis of septic patients [7,8].

Following cardiac surgery, patients usually present an elevation of some biomarkers such as procalcitonin (PCT), C-reactive protein (CRP) and leukocytes due to an acute inflammatory response [9,10]. Furthermore, it has been reported that 34% of acute kidney injury (AKI) is related to major surgery [11]. In particular, cardiac surgery leads to a well-established risk of renal dysfunction [12]. A decreased renal function has been related to a higher cardiac morbidity and mortality in patients undergoing this type of surgery [13,14]. It has been described that renal function markedly influenced some biomarkers levels such as PCT in both infected and non-infected patients after major surgery [6]. Furthermore, high levels of C-Reactive Protein have been identified as a biomarker of AKI or mortality in several clinical settings [15,16]. In addition, leukocyte count is related to AKI in patients who underwent isolated coronary artery bypass grafting with cardiopulmonary bypass [17] and neutrophil count has been described as a marker of AKI in different diseases [18,19,20,21,22,23]. In line with this, the need to use different thresholds depending on the presence/absence of renal failure has been addressed in the case of PCT in postoperative infection. Amour J et al. [6] evaluated 276 patients to determine whether the accuracy of PCT in diagnosing postoperative infection is affected by renal function after vascular surgery [5]. Regarding cardiac surgery, Jebali et al. [6] concluded that PCT is a valuable marker of bacterial infections after cardiac surgery; however, to date, there are no studies that have evaluated the influence of impaired renal function on its diagnostic accuracy in the postoperative period of this type of surgery. 

In this sense, the objective of this study is to evaluate the differential behaviour of PCT as well as its usefulness in the diagnosis of postoperative pneumonia, based on the presence or absence of deterioration of renal function in patients after cardiac surgery.

## 2. Materials and Methods

### 2.1. Patient Selection

A total of 805 adult patients who underwent heart valve surgery with cardiopulmonary bypass (CPB) and admitted at the “Hospital Clínico Universitario de Valladolid” (Valladolid, Spain) were prospectively recruited between June 2012 and January 2016. Patients were followed-up until June 2016. Patients <18 years, with a recent medical history of coronary artery disease, coronary artery bypass grafting, heart transplantation, with preoperative paced rhythms and those patients who required acute dialysis were excluded. As a second part of the initial study [24], patients were split into two groups depending on the presence of postoperative bacterial pneumonia. Centers for Disease Control and Prevention definitions for pulmonary infection were used [25]. The techniques and the treatment received by patients in the intensive care unit (ICU) did not differ from ordinary procedures. The study was approved by the Hospital’s Clinical Ethics Committee (CEIm) and informed consent was obtained from all study participants. This study followed the code of ethics of the World Medical Association (Declaration of Helsinki). 

### 2.2. Definition of Impairment in Renal Function

Creatinine was measured at ICU admission and at 8, 16, 24, 48 and 72 h after surgery. Creatinine clearance (CCr), which is a widely used test to estimate the glomerular filtration rate and is defined as the volume of plasma which is completely cleared of creatinine within a unit of time, was estimated using the Cockcroft formula [12]. Thus, an impairment in renal function was defined as the presence of a postoperative CCr <50 mL/min at 16 h after surgery which was maintained for at least 48 h.

### 2.3. Procalcitonin and C-Reactive Protein Quantification

Procalcitonin measurement in plasma was performed by electrochemiluminescence immunoassay on a chemistry analyser (Cobas 6000, Roche Diagnostics, Meylan, France) with a limit of detection of 0.02 ng/mL. Serum C-reactive protein was measured by particle enhanced immunoturbidimetric assay (e501 Module Analyzer, Roche Diagnostics) with a limit of detection of 0.15 mg/dL. The determinations were made on admission to the post-surgical critical care unit and during their stay at 8, 16, 24 and 72 h.

### 2.4. Statistical Analysis

All data were analysed using the IBM SPSS 22.0 software (SPSS, Chicago, IL) and R version 3.0.1 (R Foundation for Statistical Computing, Vienna, Austria). Data are expressed as mean ± SD or median (95% confidence interval) in non-normally distributed variables. Differences between groups were assessed using the χ^2^ test for categorical variables and the Mann–Whitney U test for continuous variables. 

Assessment of diagnostic accuracy was performed by calculating the sensitivity, specificity, positive and negative predictive values and accuracy (defined as the sum of concordant cells divided by the sum of all cells in the two-by-two table) and their 95% confidence interval. We determined the receiver operating characteristic (ROC) curve and calculated the area under the ROC curve as well as its 95% confidence interval. The accuracy of the test depends on how well the test separates the group being tested and is reflected by the area under the ROC curve. Comparison of areas under the ROC curve was performed using a nonparametric technique. The ROC curve was used to determine the optimal threshold for PCT to diagnose infection. The optimal threshold was the one that minimised the distance to the ideal point (sensitivity = specificity = 1) on the ROC curve. 

Multivariate logistic regression analyses over time adjusted by age and sex were performed to evaluate the association between inflammatory biomarkers and an impairment of renal function. The generalised linear mixed model (GLMM), taking time into account and considering each individual as a random factor, was employed to explore the effect on creatinine clearance of all inflammatory biomarkers. We considered 2-sided *p*-values < 0.05 to indicate statistical significance. 

## 3. Results

### 3.1. Clinical Characteristics

A total of 42 patients presented pulmonary infection, while 761 patients did not. Baseline characteristics of patients are reported in Table 1. Patients were similar in terms of age and sex. Patients in the infection group more frequently presented a clinical history of diabetes mellitus (33.3 (14) vs. 19.4 (148), *p* = 0.029), stroke (14.3 (6) vs. 4.2 (32), *p* = 0.003) and atrial fibrillation (64.3 (27) vs. 35 (266), *p* < 0.001). As expected, higher organ dysfunction evaluated by the SOFA score was present in infected patients (10 (3.50) vs. 3 (3), *p* < 0.001). This group of patients presented higher levels of Troponin T (1129.50 (2640.10) vs. 520.90 (449.30), *p* < 0.001), PCT (0.51 (1.21) vs. 0.34 (0.47), *p* = 0.014), creatinine (1.34 (0.94) vs. 0.95 (0.45), *p* < 0.001), glucose (188.35 (79.2) vs. 167.2 (44), *p*= 0.002), GOT (92.95 (139.6) vs. 51.55 (34.4), *p* = 0.002), LDH (490 (390) vs. 355.5 (141), *p* < 0.001), CK-MB (39.25 (62.08) vs. 22.24 (26.77), *p* < 0.001) and total bilirubin (1.20 (1.70) vs. 0.80 (0.77), *p* < 0.001), as well as higher levels of white blood cells (13925 (7717.5) vs. 11565 (4857.5), *p* = 0.003) and neutrophils (11535 (7124.14) vs. 9822.19 (4404.18), *p* = 0.002) in comparison with the no-infection group. Patients with infection also showed higher hospital mortality (61.9 (26) vs. 3.2 (24), *p* < 0.001) just as prolonged hospital stay (20.5 (21) vs. 13 (8), *p* < 0.001), ICU stay (11 (13) vs. 3 (3), *p* < 0.001) and mechanical ventilation length (120 (171) vs. 6 (3), *p* < 0.001). 

### 3.2. Biomarker Levels Based on Presence/Absence of Renal Failure and over Time in Patients with Infection

As shown in Table 2, as a result of the comparison between the groups of patients with and without renal insufficiency within the infected patients, it was observed that there were only statistically significant differences in PCT levels upon admission to the ICU, being significantly higher in patients with renal insufficiency (0.30 (0.25) vs. 0.15 (0.13), *p* = 0.021). In the infected patients, the rest of the PCT measurements throughout their stay in the ICU were not influenced by the presence or absence of impaired renal function.

In Figure 1, box plots show biomarker levels over time in infected patients based on presence/absence of renal failure. They show the progressive increase in CRP levels in infected patients, both with and without kidney failure, as well as PCT, reaching their highest value at 72 h.

### 3.3. Biomarker Levels Based on the Presence/Absence of Renal Failure and over Time in Patients without Infection

This comparison revealed that PCT levels, white blood cells and neutrophil counts at 16 h and 24 h were significantly higher in non-infected patients with renal failure. Likewise, PCT and C-Reactive protein levels were significantly higher in this group of patients at 48 h and 72 h (Table 3). In Figure 2, box plots show biomarker levels over time in non-infected patients based on the presence/absence of renal failure. 

### 3.4. Biomarkers Association with Presence of Renal Failure over Time in Patients with Infection

The generalised linear mixed model (GLMM) revealed a significant effect on creatinine clearance of all variables except neutrophils. Under this model, taking time into account and considering each individual as a random factor, we observed that PCT decreases by 0.33 ng/mL for each unit that increases creatinine clearance (Table 4). 

### 3.5. Biomarker Association with Presence of Renal Failure over Time in Patients without Infection

The generalised linear mixed model (GLMM) revealed a significant effect on creatinine clearance of neutrophils alone. Under this model, taking time into account and considering each individual as a random factor, we observed that neutrophils decrease by 0.04 cells/mL for each unit that increases creatinine clearance (Table 5). 

The area under the ROC curve of PCT in patients without kidney failure was 0.70 (95% CI 0.52–0.88), while it was 0.74 (95% CI 0.63–0.85) for patients with kidney failure (Figure 3). The optimal threshold of the PCT value for the diagnosis of infection in patients without renal failure was 0.78 ng/mL (sensitivity of 0.6 and specificity of 0.86), while it was 1.00 ng/mL (sensitivity of 0.7 and specificity of 0.79) in patients with renal failure. When comparing the ROC curves in the two subgroups according to renal function, the diagnostic accuracy of PCT was not significantly different between these two subgroups (Figure 3), but the optimal threshold differed significantly (Table 6). In contrast, the diagnostic accuracy of PCT increased significantly when the threshold was adapted to renal function (Table 7).

## 4. Discussion

The main result of our study is that PCT is a good marker for the diagnosis of postoperative infection in cardiac surgery, even in the presence of renal dysfunction. 

We found that PCT increased significantly in patients with postoperative pneumonia after cardiac surgery, in agreement with previous studies [9]. It has been previously postulated that PCT is superior to commonly used laboratory tests such as CRP or white blood cell count, and even correlates with the severity of microbial infection. The superiority of PCT can be explained by its more specific increase in bacterial infection, but also by its perioperative kinetics after CBP. In fact, in the control group without infection PCT increased slightly and transiently after CBP, while the increase in CRP was longer (Figure 2). This confirms the same finding as in the study by Jebali et al. [9], indicating that PCT is useful in the early recognition of infection after CBP.

After major surgery, the activation of inflammatory cascades can occur, which shows similarities to those seen in infectious processes. This is especially true after cardiac surgery [26], since CBP, along with ischemia and reperfusion, triggers the activation of inflammatory cascades, as well as increased thrombotic risk and higher levels of fibrinogen, catecholamines and stress hormones [27]. In this context, after CBP, the cut-off value of PCT as a marker of infection remains a matter of debate. Our results reveal an optimal PCT threshold for the diagnosis of postoperative pulmonary infection in cardiac surgery of 0.78 ng/mL, with a sensitivity of 0.6 and a specificity of 0.86. Aouifi et al. [28] demonstrated that PCT was reliable for the diagnosis of infection after cardiac surgery in the presence of fever, with a sensitivity and specificity of 0.85 and 0.95, respectively, and with a cut-off value of 1 ng/mL. Using a cut-off value of 0.5 ng/mL, Al Nawas et al. [29] found a sensitivity and specificity of 60% and 79%, respectively. 

Regarding the influence of renal function on PCT levels, we observed that its levels are higher in the presence of renal dysfunction. The generalised linear mixed model (GLMM) in infected patients revealed a significant effect of creatinine clearance on PCT, observing decreases in PCT of 0.33 ng/mL for each unit that increases creatinine clearance, which confirms the influence of renal dysfunction on this biomarker. We calculated the optimal PCT threshold for the diagnosis of postoperative infection in patients with renal dysfunction at 1.00 ng/mL, with a sensitivity of 0.7 and a specificity of 0.79 (Figure 3), also finding that the diagnostic accuracy of PCT increased significantly when the PCT threshold was adapted to renal function (Table 7). These results are comparable to those found by Amour et al. [5] in vascular surgery, strongly suggesting that renal function should be considered when evaluating the accuracy of PCT in diagnosing infections in cardiac surgery.

Some limitations in our study deserve consideration. First, we did not include all types of infections in the present study; we only evaluated pneumonia, since it is the main cause of infection during the postoperative period in cardiac surgery. This could result in an underestimation of infected patients, although other infections such as those of the urinary tract are known to stimulate little PCT secretion [30]. Second, the small number of patients with postoperative infection could imply a low power to detect a significant difference in the area under the ROC curve.

## 5. Conclusions

Procalcitonin is an early and specific biological marker of pneumonia in patients undergoing cardiac surgery, both in the presence and absence of renal dysfunction. In cardiac surgery, given a high suspicion of pneumonia, values greater than 0.78 ng/mL in patients with normal renal function and 1 ng/mL in those with renal dysfunction should be considered strong predictors of infectious complications, together with the rest of the diagnostic criteria. The routine determination of procalcitonin, with an adequate interpretation adapted to the renal function of patients, can improve their postoperative treatment, allowing a faster and more specific diagnosis and avoiding the unnecessary use of antibiotics that are known to select resistant strains. 

## Figures and Tables

**Figure 1 jcm-11-07274-f001:**
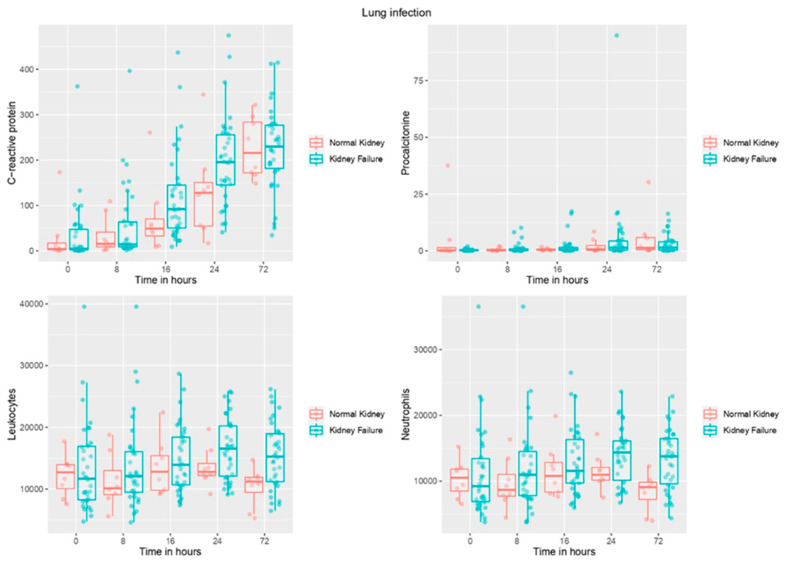
Box plots showing biomarker levels over time based on the presence/absence of renal failure in patients with infection.

**Figure 2 jcm-11-07274-f002:**
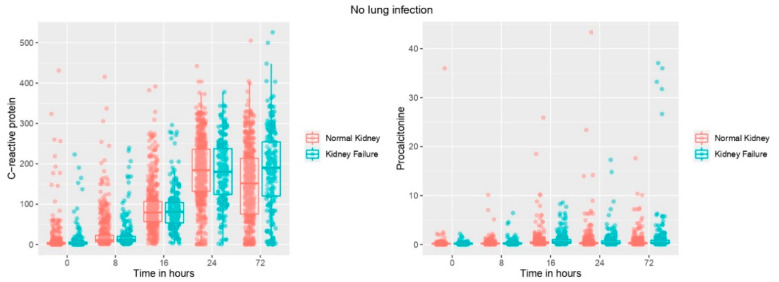
Box plots showing biomarker levels over time based on the presence/absence of renal failure in patients without infection.

**Figure 3 jcm-11-07274-f003:**
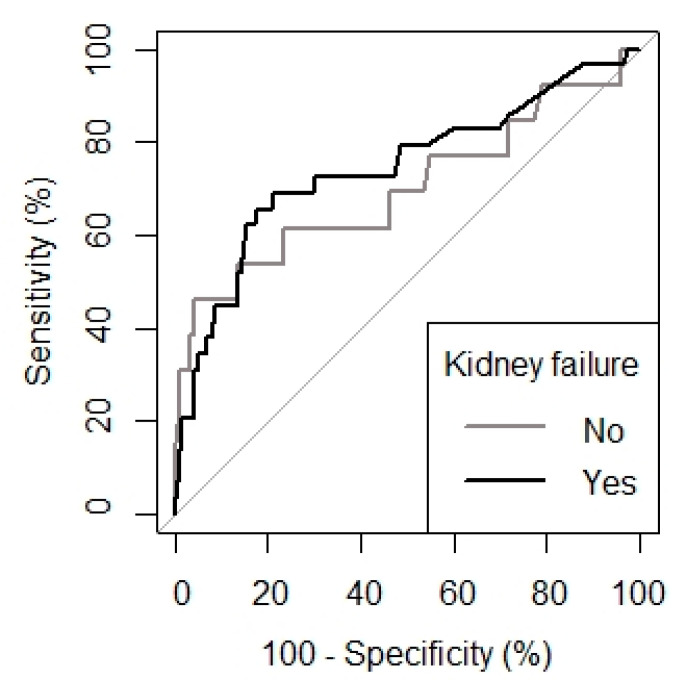
Comparison of the receiver-operating characteristic (ROC) curves in determining the predictive value of procalcitonin for the diagnosis of postoperative infection in patients without or with postoperative renal dysfunction.

**Table 1 jcm-11-07274-t001:** Characteristics of the patients in the infection and non-infection groups. Continuous variables are represented as median and interquartile range (IQR); categorical variables are represented as number (*n*) and percentages (%). ICU: intensive care unit; SOFA: Sequential Organ Failure Assessment Score; INR: international normalised ratio. *p* values < 0.05 are highlighted in bold.

	No Infection (1) (*n* = 761)	Infection (2) (*n* = 42)	*p* Value (1 vs. 2)
**Characteristics**			
Age [years, median (IQR)]	14 (70)	11 (72.50)	0.25
Male [*n*, (%)]	364 (47.8)	23 (54.8)	0.38
Functional classification (NYHA)			
I	1 (2)	1 (2)	0.07
II	70.4 (536)	59.5 (25)	0.13
III-IV	29.6 (225)	40.5 (17)	0.13
**Comorbidities, (*n*, (%))**			
Peripheral arterial disease	20 (2.6)	2 (4.8)	0.41
Chronic respiratory disease	61 (8)	3 (7.1)	0.84
High Blood Pressure	669 (87.9)	39 (92.9)	0.33
Chronic renal failure	42 (5.5)	4 (9.5)	0.28
Diabetes mellitus	148 (19.4)	14 (33.3)	**0.029**
Dyslipidaemia	557 (73.2)	35 (83.3)	0.15
Stroke	32 (4.2)	6 (14.3)	**0.003**
Chronic ischemic cardiac disease	54 (7.1)	4 (9.5)	0.55
Previous valve intervention	88 (11.6)	8 (19)	0.15
Atrial fibrillation	266 (35)	27 (64.3)	**<0.001**
EuroSCORE II, %	0.8 (1.65)	1.1 (2.06)	**0.001**
Intraoperative characteristics			
Time of CPB, min	95 (38)	122.50 (67)	**<0.001**
Time of aortic cross-clamp, min	69 (30)	91 (47)	**<0.001**
Defibrillation	165 (21.7)	10 (23.8)	0.75
**Laboratory assessments, (median (IQR))**			
Weight (kg)	74 (15)	70 (16)	0.42
Height (cm)	161 (15)	156 (19)	**0.020**
Troponin T hs (pg/mL)	520.90 (449.3)	1129.50 (2640.1)	**<0.001**
Total bilirubin (mg/dL)	0.8 (0.7)	1.20 (1.7)	**<0.001**
GOT (U/L)	51.55 (34.4)	92.95 (139.6)	**0.002**
Glucose (mg/dL)	167.2 (44)	188.35 (79.2)	**0.002**
Creatinine (mg/dL)	0.95 (0.45)	1.34 (0.9)	**<0.001**
Na (mmol/L)	138.91 (3.5)	138.48 (6.9)	0.52
K (mmol/L)	4 (0.62)	4.43 (0.7)	**<0.001**
Ca (mg/dL)	8.01 (0.5)	8 (0.7)	0.83
Mg (mg/dL)	1.66 (0.3)	1.62 (0.3)	0.43
Cl (mmol/L)	101 (4)	102 (7.6)	0.40
Urea (mg/dL)	37.9 (19.7)	56.9 (20.6)	**<0.001**
CPK (U/mL)	544 (412)	632.5 (904)	0.12
CK-MB (U/L)	22.24 (26.7)	39.25 (62.1)	**<0.001**
LDH (U/L)	355.5 (141)	490 (390)	**<0.001**
Lactate (mmol/L)	0.91 (1.5)	0.44 (1.83)	0.09
Procalcitonin (ng/mL)	0.34 (0.4)	0.51 (1.2)	**0.014**
C-Reactive Protein (mg/L)	79.79 (49.7)	82.49 (98.6)	0.72
Erythrocytes (cells/mm^3^)	3710 (620)	3675 (490)	0.52
Haematocrit (%)	32.7 (5.3)	32 (4.9)	0.37
Haemoglobin (g/dL)	10.9 (2)	10.85 (1.7)	0.24
Platelet count (cells/mm^3^)	136,000 (57000)	135,000 (59000)	0.30
White Blood cells (cells/mm^3^)	11,565 (4857.5)	13,925 (7717.5)	**0.003**
Neutrophils (cells/mm^3^)	9822.19 (4404.1)	11,535 (7124.1)	**0.002**
SOFA score	3 (3)	10 (3.50)	**<0.001**
**Outcome**			
Length of hospital stay, [days, median (IQR)]	13 (8)	20.5 (21)	**<0.001**
Length of ICU stay, [days, median (IQR)]	3 (3)	11 (13)	**<0.001**
Length of mechanical ventilation, [hours, median (IQR)]	6 (3)	120 (171)	**<0.001**
Mortality, [*n* (%)]	24 (3.2)	61.9 (26)	**<0.001**

**Table 2 jcm-11-07274-t002:** Biomarker levels based on the presence/absence of renal function impairment over time in patients with lung infection. Biomarkers are represented as median and interquartile range (IQR). *p* values < 0.05 are highlighted in bold.

	No Renal Function Impairment (*n* = 13)	Renal Function Impairment (*n* = 29)	*p* Value
Procalcitonin 0 h (ng/mL)	0.15 (0.13)	0.30 (0.25)	**0.021**
C-Reactive Protein 0 h (mg/L)	2.98 (21.80)	4.95 (53.29)	0.06
White Blood cells 0 h (cells/mm^3^)	11640 (8150)	11,860 (8745)	0.47
Neutrophils 0 h (cells/mm^3^)	9463.32 (7378.21)	10,223.32 (5991.08)	0.52
Procalcitonin 8 h (ng/mL)	0.20 (0.85)	0.30 (0.86)	0.40
C-Reactive Protein 8 h (mg/L)	10.62 (39.47)	15.19 (95.39)	0.19
White Blood cells 8 h (cells/mm^3^)	11,990 (6365)	11,280 (7530)	0.42
Neutrophils 8 h (cells/mm^3^)	10,611.15 (5698.08)	9956.25 (7014.15)	0.36
Procalcitonin 16 h (ng/mL)	0.40 (0.96)	0.90 (1.65)	0.63
C-Reactive Protein 16 h (mg/L)	89.55 (80.04)	75.43 (136.98)	0.63
White Blood cells 16 h (cells/mm^3^)	14,060 (8625)	13,820 (7565)	0.82
Neutrophils 16 h (cells/mm^3^)	11,838.52 (8051.52)	11,233.25 (6893)	0.73
Procalcitonin 24 h (ng/mL)	0.25 (2.98)	1.50 (4.40)	0.09
C-Reactive Protein 24 h (mg/L)	179.33 (148.41)	160.78 (132.95)	0.73
White Blood cells 24 h (cells/mm^3^)	13,970 (9360)	15850 (8295)	0.84
Neutrophils 24 h (cells/mm^3^)	12,838.43 (8834.61)	13,730.84 (6143.56)	0.99
Procalcitonin 48 h (ng/mL)	0.81 (2.55)	0.84 (2.42)	0.88
C-Reactive Protein 48 h (mg/L)	265.09 (127.48)	221.95 (134.19)	0.78
White Blood cells 48 h (cells/mm^3^)	14,120 (11855)	16,140 (7055)	0.48
Neutrophils 48 h (cells/mm^3^)	12,623.28 (10,918.98)	13,769.67 (5675.79)	0.67
Procalcitonin 72 h (ng/mL)	0.79 (6.17)	1.90 (3.82)	0.34
C-Reactive Protein 72 h (mg/L)	195.10 (89.37)	243.78 (104.55)	0.35
White Blood cells 72 h (cells/mm^3^)	11820 (3710)	15410 (9465)	0.12
Neutrophils 72 h (cells/mm^3^)	10,160.29 (4708.34)	13,851 (8306.30)	0.08

**Table 3 jcm-11-07274-t003:** Biomarker levels based on the presence/absence of renal function impairment over time in patients without lung infection. Biomarkers are represented as median and interquartile range (IQR). *p* values < 0.05 are highlighted in bold.

	No Renal Function Impairment (*n* = 562)	Renal Function Impairment (*n* = 199)	*p* Value
Procalcitonin 0 h (ng/mL)	0.20 (0.13)	0.19 (0.13)	0.847
C-Reactive Protein 0 h (mg/L)	1.84 (4.39)	2.37 (6.16)	0.091
White Blood cells 0 h (cells/mm^3^)	10,810 (4945)	10,360 (5550)	0.782
Neutrophils 0 h (cells/mm^3^)	8969.94 (4343.85)	8675.04 ()	0.893
Procalcitonin 8 h (ng/mL)	0.20 (0.15)	0.20 (0.20)	0.143
C-Reactive Protein 8 h (mg/L)	11.21 (14.55)	12.22 (18.71)	0.383
White Blood cells 8 h (cells/mm^3^)	11,380 (4755)	11,210 (5090)	0.824
Neutrophils 8 h (cells/mm^3^)	9624.36 (4430.27)	9459.84 (4881.14)	0.809
Procalcitonin 16 h (ng/mL)	0.30 (0.36)	0.59 (0.89)	**<0.001**
C-Reactive Protein 16 h (mg/L)	79.64 (48.31)	80.82 (57.13)	0.885
White Blood cells 16 h (cells/mm^3^)	11,430 (4645)	12,210 (5490)	**0.036**
Neutrophils 16 h (cells/mm^3^)	9713.34 (4140.28)	10,350.64 (5139.92)	**0.031**
Procalcitonin 24 h (ng/mL)	0.24 (0.24)	0.33 (0.62)	**<0.001**
C-Reactive Protein 24 h (mg/L)	182.65 (104.10)	186.61 (110.81)	0.578
White Blood cells 24 h (cells/mm^3^)	11,960 (5080)	12,860 (5800)	**0.003**
Neutrophils 24 h (cells/mm^3^)	9877 (4785.94)	11,025 (5498.64)	**0.001**
Procalcitonin 48 h (ng/mL)	0.28 (0.38)	0.51 (0.96)	**<0.001**
C-Reactive Protein 48 h (mg/L)	173.11 (134.90)	219.90 (133.24)	**<0.001**
White Blood cells 48 h (cells/mm^3^)	11,180 (5265)	11,570 (5622.50)	0.199
Neutrophils 48 h (cells/mm^3^)	8964.78 (4680.39)	9487.68 (4892.40)	0.118
Procalcitonin 72 h (ng/mL)	0.25 (0.25)	0.47 (0.82)	**<0.001**
C-Reactive Protein 72 h (mg/L)	153.80 (140.62)	188.79 (126.82)	**<0.001**
White Blood cells 72 h (cells/mm^3^)	10,270 (5155)	10,230 (5562.50)	0.517
Neutrophils 72 h (cells/mm^3^)	8113.74 (4697.66)	8364.55 (5509.24)	0.311

**Table 4 jcm-11-07274-t004:** Generalised linear mixed models (GLMM) for evaluating the association between biomarkers and creatinine clearance over time in patients with lung infection.

	Estimate	CI 95% min	CI 95% max	*p*-Value
Age	−0.9334000000	−1.59	−0.28	**0.009050**
Gender	0.8350000000	−9.82	11.49	0.88058
Time	−0.0162300000	−0.05	0.02	0.412307
CRP	−0.0378300000	−0.05	−0.03	**2.31 × 10^−10^**
PCT	−0.3307000000	−0.43	−0.23	**1.08 × 10^−10^**
Leukocytes	−0.0015920000	0.00	0.00	**0.000438**
Neutrophils	0.0008293000	0.00	0.00	0.087382

**Table 5 jcm-11-07274-t005:** Generalised linear mixed models (GLMM) for evaluating the association between biomarkers and creatinine clearance over time in patients without infection.

	Estimate	CI 95% min	CI 95% max	*p*-Value
Age	125.5000000000	78.30	172.98	7.81 × 10^−6^
Gender	−0.9157000000	−1.57	−0.27	0.01010
Time	0.9311000000	−9.58	11.43	0.865820
CRP	−0.0082450000	−0.10	0.09	0.863720
PCT	0.0005547000	0.00	0.00	0.611500
Leukocytes	−0.0011590000	0.00	0.00	0.249550
Neutrophils	−0.0419700000	−0.07	−0.02	0.00299

**Table 6 jcm-11-07274-t006:** Comparison of the AUROC of procalcitonin for the diagnosis of postoperative infection according to postoperative renal function.

	Renal Function Impairment (*n* = 228)	No Renal Function Impairment (*n* = 574)
Threshold of procalcitonin (ng/mL)	1.00	0.78
AUROC	0.74 (0.63–0.85)	0.70 (0.51–0.88)
*p*-value vs. no discrimination curve	<0.001	0.01

**Table 7 jcm-11-07274-t007:** Comparison of the efficiency of procalcitonin either with a fixed threshold or a threshold adapted to postoperative renal dysfunction.

Variable	Procalcitonin 0.78 ng/mL	Adapted Procalcitonin 1.00 ng/mL
Sensitivity	0.65 (0.47–0.80)	0.62 (0.44–0.77)
Specificity	0.67 (0.60–0.73)	0.74 (0.67–0.79)
Positive predictive value	0.23 (0.15–0.33)	0.26 (0.17–0.37)
Negative predictive value	0.93 (0.88–0.96)	0.92 (0.88–0.96)
Accuracy	0.67	0.72

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
