# Peer review of "Influence of Renal Dysfunction on the Differential Behaviour of Procalcitonin for the Diagnosis of Postoperative Infection in Cardiac Surgery"

_jcm, 2022, doi:10.3390/jcm11247274_

Round 1

Reviewer 1 Report

The authors concluded that procalcitonin is a biomarker for the diagnosis of postoperative infection in cardiac surgery, regardless of the presence of renal dysfunction. Considering renal dysfunction, the PCT threshold for predicting infection is different. Therefore, the authors proposed new renal dysfunction specific thresholds for the diagnosis of pulmonary infection. I think this is good work but not good in result delivery. My concerns are listed below,

Figure 3 used ROC to evaluate the performance of PCT in predicting postoperative infection in patients without or with postoperative renal dysfunction. How about the result of ROC for the postoperative infection, say the labels are infection and non-infection, regardless of the kidney status.

Figure 1 shows the progressive increase in CRP levels in infected patients, both with and without kidney failure. But for PCT, the trend is not that clear. Scale the y-axis and overlooking the outlier points may make it much clearer.

 In Figure 2, the box plots show biomarkers levels across time in non-infected patients based on presence/absence of renal failure. The scale of Y-axis is still a problem for represent the main result.

 The use of Procalcitonin and PCT is not consistent in the manuscript.

 Other than PCT, several works about other infection biomarkers using omics data have been proposed, such as doi: 10.1109/TCBB.2021.3107874, doi.org/10.1186/s12864-021-07576-4, and doi.org/10.1002/ctm2.123, which is recommended to introduce or discuss them.

Author Response

We are very grateful to have the opportunity to review our manuscript entitled “INFLUENCE OF RENAL DYSFUNCTION ON THE DIFFERENTIAL BEHAVIOR OF PROCALCITONIN FOR THE DIAGNOSIS OF POSTOPERATIVE INFECTION IN CARDIAC SURGERY." and we have taken into account the reviewers' notes. We want to thank the editors and reviewers for the time they have spent reading our study and their insightful guidance. After carefully considering all comments, we expect that the document is now worthy of publication in Journal of Clinical Medicine.

On the following pages we provide specific responses to each of the observations.

REVIEWER 1

The authors concluded that procalcitonin is a biomarker for the diagnosis of postoperative infection in cardiac surgery, regardless of the presence of renal dysfunction. Considering renal dysfunction, the PCT threshold for predicting infection is different. Therefore, the authors proposed new renal dysfunction specific thresholds for the diagnosis of pulmonary infection. I think this is good work but not good in result delivery. My concerns are listed below,

Figure 3 used ROC to evaluate the performance of PCT in predicting postoperative infection in patients without or with postoperative renal dysfunction. How about the result of ROC for the postoperative infection, say the labels are infection and non-infection, regardless of the kidney status.

We greatly appreciate your comment. As indicated by the reviewer, Figure 3 compares the ROC curves in patients with and without postoperative renal dysfunction to determine the predictive value of procalcitonin in the diagnosis of postoperative infection. In this study, the analysis of the ROC curves was not carried out regardless of the state of the kidney, since the objective was to analyze the behavior of the PCT influenced by renal dysfunction, finding as a result that the optimal threshold of the PCT value for the diagnosis of infection in patients without renal insufficiency was 0.78 ng/ml, while in patients with renal insufficiency it was 1.00 ng/ml. Thus, the adaptation of the threshold in these patients would improve diagnostic accuracy.

Figure 1 shows the progressive increase in CRP levels in infected patients, both with and without kidney failure. But for PCT, the trend is not that clear. Scale the y-axis and overlooking the outlier points may make it much clearer.

Thank you very much for your comment. Giving value to your consideration, we have tried to scale the y-axis of figure 1 to improve its compression, but the visual result obtained has been similar, and since the measurements of the infection markers were taken at 8, 16, 24 and 48 hours after surgery, we finally considered keeping this same scale on the y-axis for a better interpretation of the results.

 In Figure 2, the box plots show biomarkers levels across time in non-infected patients based on presence/absence of renal failure. The scale of Y-axis is still a problem for represent the main result.

We greatly appreciate your suggestion to improve our work. In the same way that we have done in figure 1, we have tried to improve the scale of the axis and in figure 2, but, since in both cases the visual result does not differ from the current one, we have considered keeping the same scale in both figures to also maintain the correlation with the times of sample collection and biomarker analysis.

 The use of Procalcitonin and PCT is not consistent in the manuscript.

We are very grateful for this annotation and following the instructions of the reviewer we have made the changes indicated in the lines 64, 70, 71, 73, 77, 123, 141, 212, 218, 220, 235, 237, 241, 242, 244, 251, 252, 254, 263, 265

 Other than PCT, several works about other infection biomarkers using omics data have been proposed, such as doi: 10.1109/TCBB.2021.3107874, doi.org/10.1186/s12864-021-07576-4, and doi.org/10.1002/ctm2.123, which is recommended to introduce or discuss them.

Thanks for the comment, we agree that, since infection is an important complication with potentially devastating consequences, it is important to be aware of the growing number of molecular research papers, so we have added these studies between lines 57-59 of our manuscript.

Reviewer 2 Report

Major comments

How did you define (? bacterial) pneumonia? Did you try assessing the impact of its severity and distribution, presence of bacteremia, etc?

Renal failure has been determined as a eGFR<50 by 16 h from surgery and throughout the subsequent 32 h. This definition lacks dynamics in SCr and extrapolated eGFR along time, in other words it tell us nothing about the delta in these parameters – the nature of renal functional impairment: is it a stable CKD, evolving AKI or their combination. Conceivably, the kinetics of procalcitonin differ in these three situations. In these perspectives, the analysis of biomarkers not sub-classified accordingly might be meaningless looking at "kidney failure" as a homogenous group.

Likely an AUC for creatinine or eGFR within a pre-specified post-operative period would give a closer estimate of the impact of renal function upon PCT levels, perhaps also presented as an EUC for the corresponding time period.  

Furthermore, I cannot fully understand what data is presented: Lab results on Table 1 – what does "baseline" means? Before operation? By 16h?  

Table 2 and Figure 1- Lab results – why not adding SCr and eGFR at all time points? This would provide at least some insight regarding the dynamics in kidney function, addressed above

Table 2  - It took me time to understand that the numbers presented on top (13 and 29) are related only to patients with pneumonia, you should better emphasize this point, also substituting "infection" with "lung infection" or "pneumonia".

Same comment for the headings of Table 3.   

Tables 4 and 5 – I believe I understand that the data generated in these tables reflect the variables at all time points for all patients grouped together separately for patients with and without lung infection. My question, in line with the comments above, is: Are all imputed values corrected for their corresponding eGFR for each specific time point?

Were patients requiring acute dialyses included in the study?

On my opinion, the bottom line is that the practical usefulness of PCT for the detection of evolving lung infection following cardiac surgery remains questionable, in the presence or absence of renal dysfunction, taking into account the huge overlap of PCT values in patients with- and without infection. This is despite your ability to show that correcting the threshold value for eGFR improves its predictive value. Would you frankly decide upon antibiotics, or avoid antibiotics, solely based on PCT levels upon admission to the ICU, corrected for eGFR? I believe this conclusion deserves emphasize in the Abstract and Conclusions.   

Minor comments

Table 1, data presentation is confusing. Usually, the number of patients should appear first, followed by percentage in parenthesis. Consider switching the order accordingly, and provide the order of data presented in "Characteristics", as you did for Comorbidities"  

Table 1 - the variable "Functional classification (NYHA)" should include a number. I assume it should be I, whereas the line bellow should be II and not I-II   

Inaccurate numbering of references in line 73

The paragraph beginning at line 158 – I suggest starting by  " As shown in Table 2…"

Author Response

We are very grateful to have the opportunity to review our manuscript entitled “INFLUENCE OF RENAL DYSFUNCTION ON THE DIFFERENTIAL BEHAVIOR OF PROCALCITONIN FOR THE DIAGNOSIS OF POSTOPERATIVE INFECTION IN CARDIAC SURGERY." and we have taken into account the reviewers' notes. We want to thank the editors and reviewers for the time they have spent reading our study and their insightful guidance. After carefully considering all comments, we expect that the document is now worthy of publication in Journal of Clinical Medicine.

On the following pages we provide specific responses to each of the observations.

REVIEWER 2

How did you define (? bacterial) pneumonia? Did you try assessing the impact of its severity and distribution, presence of bacteremia, etc?

Thank you very much for allowing us the clarification. Centers for Disease Control and ORevention definitions for pulmonary infection were used. We have clarified this point and added the information to the material and methods section, line 93. Regarding bacteremia, it was not recorded, only infection was evaluated.

Renal failure has been determined as a eGFR<50 by 16 h from surgery and throughout the subsequent 32 h. This definition lacks dynamics in SCr and extrapolated eGFR along time, in other words it tell us nothing about the delta in these parameters – the nature of renal functional impairment: is it a stable CKD, evolving AKI or their combination. Conceivably, the kinetics of procalcitonin differ in these three situations. In these perspectives, the analysis of biomarkers not sub-classified accordingly might be meaningless looking at "kidney failure" as a homogenous group.

We gladly accept the reviewer's comment and agree that it would be very interesting to know the dynamics of the evolution of renal failure. In contrast, as indicated by the reviewer, in this first study to assess the influence of renal function on PCT, patients were divided into only two groups, with and without renal failure (according to the presence of a postoperative CrC < 50 ml/min at 16 h after surgery and maintained for at least 48 h). After the results obtained, the study of the kinetics of procalcitonin according to the evolution of renal function would be an interesting objective to analyze in future research.

Likely an AUC for creatinine or eGFR within a pre-specified post-operative period would give a closer estimate of the impact of renal function upon PCT levels, perhaps also presented as an EUC for the corresponding time period.  

We greatly appreciate the comment and, based on the results found in this investigation, we will evaluate the possibility of studying this point as indicated by the reviewer to offer a closer estimate of the impact of renal function on PTC levels.

Furthermore, I cannot fully understand what data is presented: Lab results on Table 1 – what does "baseline" means? Before operation? By 16h?  

Thank you very much for allowing us the clarification. "Baseline Patient Characteristics" in table 1 refers to the medical history of patients. We have modified the wording by "Characteristics of the patients in the infection and non-infection groups"

Table 2 and Figure 1- Lab results – why not adding SCr and eGFR at all time points? This would provide at least some insight regarding the dynamics in kidney function, addressed above

We greatly appreciate your suggestion to improve our work. We agree that it would be interesting to know the dynamics of SCr and eGFR at each moment, but given the complexity and the high number of groups that this analysis would generate, we opted for a primary division of the patients into two groups according to the presence or absence of renal failure (according to the criteria of the postoperative CCr level at 16 h after surgery and its permanence for at least 48 h). A more extended analysis could be the objective of future studies.

Table 2  - It took me time to understand that the numbers presented on top (13 and 29) are related only to patients with pneumonia, you should better emphasize this point, also substituting "infection" with "lung infection" or "pneumonia".

We are very grateful for this annotation and following the reviewer's instructions we have made the changes indicated in the heading of Table 2, changing "infection" to "lung infection".

Same comment for the headings of Table 3.   

In the same way, we have made the changes indicated in the heading of Table 3, changing "infection" to "lung infection".

Tables 4 and 5 – I believe I understand that the data generated in these tables reflect the variables at all time points for all patients grouped together separately for patients with and without lung infection. My question, in line with the comments above, is: Are all imputed values corrected for their corresponding eGFR for each specific time point?

Thank you for allowing us the clarification. Tables 4 and 5 correspond to the results of the generalized linear mixed model (GLMM), whose results revealed the effect of creatinine clearance on the biomarkers studied. Thus, in patients with pulmonary infection (Table 4) we observed decreases in PCT of 0.33 ng/ml for each unit that increases creatinine clearance, while in patients without pulmonary infection (Table 5) what is revealed to us are decreases of neutrophils of 0.04 cells. /ml for each unit that increases creatinine clearance.

Were patients requiring acute dialyses included in the study?

Thank you very much for allowing us to clarify this point. Patients requiring acute dialysis were excluded from this study. We have clarified this point and added this exclusion criteria in the material and methods section on line 91.

On my opinion, the bottom line is that the practical usefulness of PCT for the detection of evolving lung infection following cardiac surgery remains questionable, in the presence or absence of renal dysfunction, taking into account the huge overlap of PCT values in patients with- and without infection. This is despite your ability to show that correcting the threshold value for eGFR improves its predictive value. Would you frankly decide upon antibiotics, or avoid antibiotics, solely based on PCT levels upon admission to the ICU, corrected for eGFR? I believe this conclusion deserves emphasize in the Abstract and Conclusions.   

Thank you very much for allowing us to clarify this point. As indicated by the reviewer, the results of our study suggest that procalcitonin is a valid marker in the diagnosis of pneumonia both in the presence and absence of renal dysfunction, increasing its predictive capacity if it is adapted depending on the renal function of the patients. Despite this, it must be taken into account that it should not be used as an isolated marker in the diagnosis of infection, but rather as another tool that must be contextualized within the patient's clinical situation. Our study suggests that in the presence of a high suspicion of pulmonary infection, procalcitonin levels adapted to renal function are a reliable biological marker to take into account, together with the rest of the diagnostic criteria for pneumonia. We have clarified this point in the conclusion: “In cardiac surgery, given a high suspicion of pneumonia, values ​​greater than 0.78 ng/ml in patients with normal renal function and 1 ng/ml in those with renal dysfunction should be considered strong predictors of infectious complications, together with the rest of the diagnostic criteria.”

Minor comments

Table 1, data presentation is confusing. Usually, the number of patients should appear first, followed by percentage in parenthesis. Consider switching the order accordingly, and provide the order of data presented in "Characteristics", as you did for Comorbidities"  

Thanks for your comment. We agree that the order of the number of patients and the percentage can be confusing, we have made the change in the table following your instructions.

Table 1 - the variable "Functional classification (NYHA)" should include a number. I assume it should be I, whereas the line bellow should be II and not I-II   

We greatly appreciate your comment, and as you indicate, we have made the changes in the table to solve the error.

Inaccurate numbering of references in line 73

Thank you very much for the comment. We have modified the numbering of the references cited by the reviewer.

The paragraph beginning at line 158 – I suggest starting by  " As shown in Table 2…"

After considering the reviewer's observation to be very correct, the consequent modifications are made for a better understanding of the sentence
